# Membrane-bound *O*-acyltransferase 7 (MBOAT7) shapes lysosomal lipid homeostasis and function to control alcohol-associated liver injury

Venkateshwari Varadharajan[1,2,3], Iyappan Ramachandiran[1,2,3], William J Massey[1,2,3], Raghav Jain[4], Rakhee Banerjee[1,2,3], Anthony J Horak[1,2,3], Megan R McMullen[3,5], Emily Huang[3,5], Annette Bellar[3], Shuhui W Lorkowski[6], Kailash Gulshan[7], Robert N Helsley[1,8], Isabella James[4], Vai Pathak[3,5], Jaividhya Dasarathy[3,9], Nicole Welch[3,5], Srinivasan Dasarathy[2,3,5], David Streem[10], Ofer Reizes[2], Daniela S Allende[3,11], Jonathan D Smith[1], Judith Simcox[4], Laura E Nagy[2,3,5], J Mark Brown[2,3]*

[1]Department of Cancer Biology, Lerner Research Institute of the Cleveland Clinic, Cleveland, United States; [2]Center for Microbiome and Human Health, Lerner Research Institute, Cleveland Clinic, Cleveland, United States; [3]Northern Ohio Alcohol Center (NOAC), Lerner Research Institute, Cleveland Clinic, Cleveland, United States; [4]Department of Biochemistry, University of Wisconsin-Madison, Madison, United States; [5]Department of Inflammation and Immunity, Lerner Research Institute, Cleveland Clinic, Cleveland, United States; [6]Department of Cardiovascular and Metabolic Sciences, Lerner Research Institute of the Cleveland Clinic, Cleveland, United States; [7]Center for Gene Regulation in Health and Disease (GRHD), Cleveland State University, Cleveland, United States; [8]Department of Pharmacology & Nutritional Sciences, Saha Cardiovascular Research Center, University of Kentucky College of Medicine, Lexington, United States; [9]Department of Family Medicine, Metro Health Medical Center, Case Western Reserve University, Cleveland, United States; [10]Lutheran Hospital, Cleveland Clinic, Cleveland, United States; [11]Department of Anatomical Pathology, Cleveland Clinic, Cleveland, United States

*For correspondence:
brownm5@ccf.org

**Abstract** Recent genome-wide association studies (GWAS) have identified a link between single-nucleotide polymorphisms (SNPs) near the MBOAT7 gene and advanced liver diseases. Specifically, the common MBOAT7 variant (rs641738) associated with reduced MBOAT7 expression is implicated in non-alcoholic fatty liver disease (NAFLD), alcohol-associated liver disease (ALD), and liver fibrosis. However, the precise mechanism underlying MBOAT7-driven liver disease progression remains elusive. Previously, we identified MBOAT7-driven acylation of lysophosphatidylinositol lipids as key mechanism suppressing the progression of NAFLD (Gwag et al., 2019). Here, we show that MBOAT7 loss of function promotes ALD via reorganization of lysosomal lipid homeostasis. Circulating levels of MBOAT7 metabolic products are significantly reduced in heavy drinkers compared to healthy controls. Hepatocyte- (*Mboat7*-HSKO), but not myeloid-specific (*Mboat7*-MSKO), deletion of *Mboat7* exacerbates ethanol-induced liver injury. Lipidomic profiling reveals a reorganization of the hepatic lipidome in *Mboat7*-HSKO mice, characterized by increased endosomal/lysosomal lipids. Ethanol-exposed *Mboat7*-HSKO mice exhibit dysregulated autophagic flux and lysosomal biogenesis, associated with impaired transcription factor EB-mediated lysosomal biogenesis and autophagosome accumulation. This study provides mechanistic insights into how MBOAT7 influences

ALD progression through dysregulation of lysosomal biogenesis and autophagic flux, highlighting hepatocyte-specific MBOAT7 loss as a key driver of ethanol-induced liver injury.

## eLife assessment

Varadharajan et al. explore the mechanistic basis of MBOAT7 SNP association with steatotic liver disease and link its function in LPI acylation to altered lipidomics of endosomal/lysosomal system and impaired TFEB mediated lysosomal biogenesis. The findings are **important** with theoretical and practical implications in MAFLD, alcohol-induced hepatic steatosis, and lysosomal diseases. The strength of evidence is **convincing** using methodology in line with current state-of-the-art.

## Introduction

End-stage liver diseases account for approximately 2 million deaths annually worldwide, with nearly half of liver disease-associated deaths arising from complications of alcohol-associated and non-alcoholic fatty liver disease (NAFLD)-related cirrhosis, and the other half driven by viral hepatitis and hepatocellular carcinoma. It is generally appreciated that there are some shared mechanisms driving liver injury from viral, NAFLD, or alcohol-associated liver disease (ALD)-driven etiologies, but also etiology-specific drivers that uniquely shape the pathogenesis of liver failure. Although there has been great progress in identifying the 'multiple hits' that lead to end-stage liver disease, we are only beginning to understand the cellular and molecular mechanisms driving etiology-specific liver disease progression. Within the evolving 'multiple hit' theory of liver disease progression, it is clear that interactions between environmental factors (i.e., diet, microbiome, alcohol, viral infection, environmental toxins, etc.) and genetic determinants uniquely contribute to liver injury (*Cohen et al., 2011*; *Rinella and Sanyal, 2016*). Currently, the only option for end-stage liver disease is liver transplantation. However, the availability of viable donor livers is finite, and pharmacological approaches to improve outcomes are simply lacking due to our poor understanding of the underlying mechanisms of disease pathogenesis. Given this, there is a clear need to understand the genetic and environmental interactions promoting the progression of liver disease from simple steatosis to more advanced inflammatory and fibrotic disease. We address this gap here by investigating the mechanisms linking a recently identified liver disease susceptibility gene in combination with alcohol exposure.

Genome-wide association studies (GWAS) provide a powerful unbiased tool to identify new genes contributing to human disease, allowing for pinpoint accuracy in identification of new potential drug targets. This is exemplified by the recent success story of GWAS discoveries leading to rapid development of monoclonal antibodies targeting proprotein convertase subtilisin/kexin type 9 (PCSK9) for hyperlipidemia and cardiovascular diseases (*Sabatine, 2019*). Since 2015, several independent GWAS studies have identified a liver disease susceptibility locus (rs641738) near the genes encoding *MBOAT7* and *TMC4* (*Buch et al., 2015*; *Mancina et al., 2016*; *Krawczyk et al., 2017*; *Thabet et al., 2016*; *Thabet et al., 2017*; *Teo et al., 2021*). It is important to note that the rs641738 T-allele (~43% allele frequency in European ancestry populations) is associated with all major forms of liver injury including NAFLD, ALD, and viral hepatitis-induced fibrosis (*Buch et al., 2015*; *Mancina et al., 2016*; *Krawczyk et al., 2017*; *Thabet et al., 2016*; *Thabet et al., 2017*; *Teo et al., 2021*). The rs641738 variant is associated with a C>T missense single-nucleotide polymorphism (SNP) within the first exon the *TMC4* gene, but the GTEx project shows that *TMC4* is not abundantly expressed in human liver (1.4 transcripts per million). We previously showed that mice lacking *Tmc4* ($Tmc4^{-/-}$) have normal high fat diet-induced hepatic steatosis (*Gwag et al., 2019*). We also recently demonstrated that antisense oligonucleotide (ASO)-mediated knockdown of *Mboat7* promotes insulin resistance, hepatic steatosis, hepatocyte death, inflammation, and early fibrosis in high fat diet-fed mice (*Gwag et al., 2019*). In parallel, four independent groups also showed that *Mboat7* loss of function promotes hepatic steatosis, inflammation, and fibrosis in mice (*Meroni et al., 2020*; *Tanaka et al., 2021*; *Thangapandi et al., 2021*; *Xia et al., 2021*). Collectively, *MBOAT7* is a genetic determinant of advanced liver disease, but how this gene shapes susceptibility to environmental cues is still an area of intense investigation.

The *MBOAT7* gene encodes a lysophospholipid acyltransferase enzyme (also known as lysophosphatidylinositol acyltransferase 1, LPIAT1), which uniquely contributes to the Land's cycle of

membrane phospholipid remodeling (*Shindou and Shimizu, 2009*). The Land's cycle is a series of phospholipase-driven deacylation and lysophospholipid acyltransferase-driven acylation reactions that shape membrane asymmetry and diversity (*Shindou and Shimizu, 2009*). It is important to note that MBOAT7 selectively diversifies the fatty acid composition of membrane phosphatidylinositol (PI) species and not phospholipids with other head groups and exhibits acyl chain specificity for polyunsaturated fatty acids (*Shindou and Shimizu, 2009*; *Gijón et al., 2008*; *Zarini et al., 2014*). This substrate specificity has been observed in in vitro or cell-based studies (*Gijón et al., 2008*; *Zarini et al., 2014*), which has been confirmed in vivo in mice with diminished *Mboat7* function (*Gwag et al., 2019*; *Meroni et al., 2020*; *Tanaka et al., 2021*; *Thangapandi et al., 2021*; *Xia et al., 2021*). Although MBOAT7 is well documented to directly modulate PI lipids, the Land's cycle is highly dynamic and has the potential to influence many downstream metabolic processes as well as cell signaling. Here, we report that ethanol-induced perturbation of the hepatic lipidome is powerfully shaped by MBOAT7 function in hepatocytes. This MBOAT7-dependent reorganization of the hepatic lipidome in response to ethanol is also functionally tied to diminished lysosome function and defective autophagy. This work shows that MBOAT7 uniquely contributes to ethanol-induced liver injury via perturbations of hepatic lipid metabolism that extend beyond the direct remodeling of membrane PI.

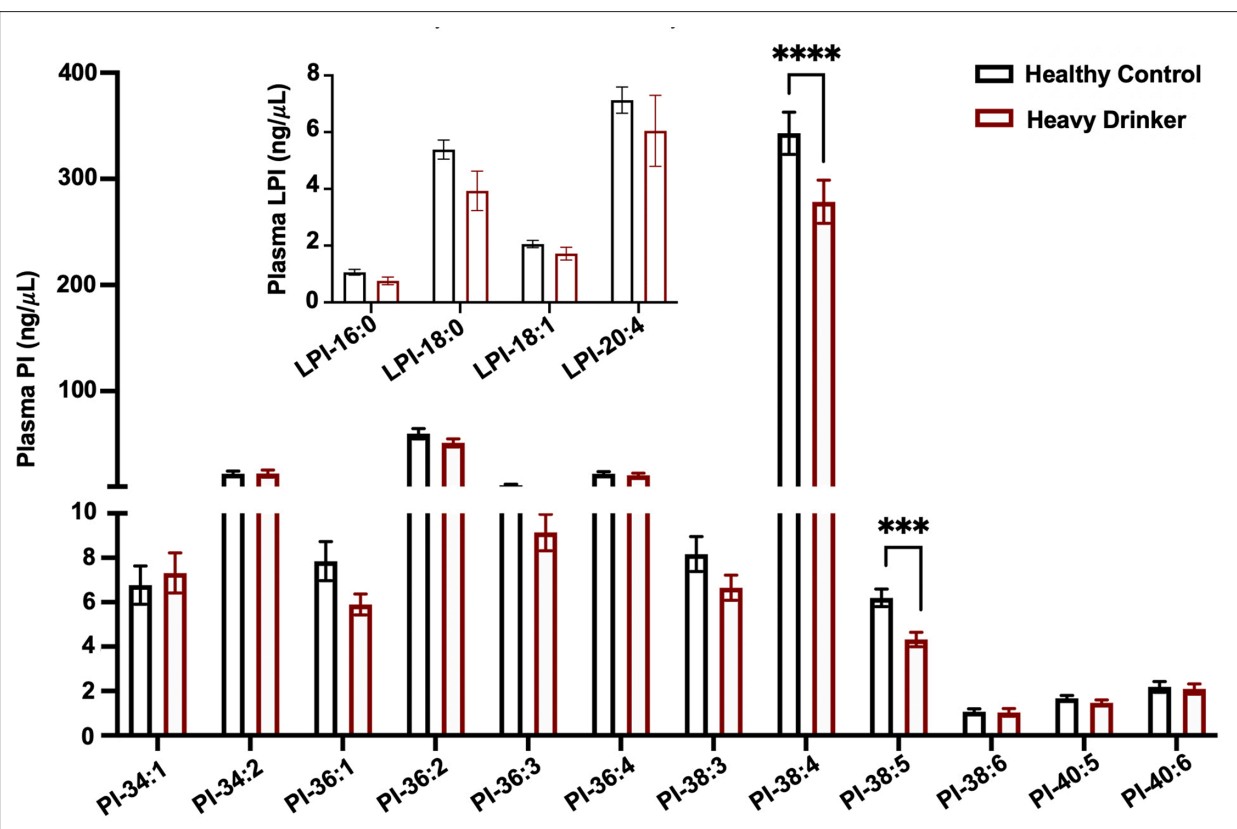

**Figure 1.** MBOAT7 products are selectively reduced in heavy drinkers. Plasma lysophosphatidylinositol (LPI – inset graph) and phosphatidylinositol (PI) species from both male and female healthy controls and heavy drinkers were measured by liquid chromatography–tandem mass spectrometry (LC–MS/MS). $n$ = 10–16; ***$p < 0.001$ and ****$p < 0.0001$ in the figure. Analysis of variance (ANOVA) with Tukey's post hoc test.

The online version of this article includes the following source data for figure 1:

**Source data 1.** Demographic and clinical parameters for the entire cohort of healthy controls and heavy drinkers recruited for this study.

# Results

## Heavy drinkers have reduced circulating levels of MBOAT7 enzymatic products

Given previous studies have shown that MBOAT7 is a risk locus for alcohol-associated cirrhosis (**Buch et al., 2015**), we investigated whether active alcohol consumption was associated with alterations in MBOAT7 function. To address this, we measured both lysophosphatidylinositol (LPI) substrates and PI products of the MBOAT7 enzymatic reaction in the circulation of healthy controls compared to confirmed heavy drinkers (**Figure 1**). Heavy drinkers were recruited and defined by an AUDIT score (**Fleming et al., 1991**) greater than 16, and compared to an age- and sex-matched healthy control population (**Figure 1—source data 1**). In agreement with genetic studies linking MBOAT7 variants to alcohol-associated cirrhosis (**Buch et al., 2015**), we find that circulating levels of metabolic products of MBOAT7 including arachidonic acid- and eicosapentaenoic acid-containing phophosphatidylinositols (PI 38:4 and PI 38:5) are significantly reduced in heavy drinkers compared to age-matched healthy controls (**Figure 1**). Given MBOAT7 demonstrates specificity for polyunsaturated (PUFA) acyl-CoA substrates (**Gijón et al., 2008**), it is important to note that only select PUFA-containing MBOAT7 products (PI 38:4 and PI 38:5) were reduced in heavy drinkers, whereas all other molecular species of PI were unaltered. We also examined the circulating levels LPI substrates of MBOAT7 but found no significant differences between controls and heavy drinkers (**Figure 1**). These data show that excessive alcoholic intake is associated with reduced levels of MBOAT7 product lipids, which further bolsters the concept that MBOAT7 loss of function may be causally linked to ALD progression.

## MBOAT7 loss of function in hepatocytes, but not myeloid cells, facilitates ethanol-induced liver injury in mice

Although there is some emerging evidence that MBOAT7 genetic variants may predispose humans to alcohol-induced liver injury (**Buch et al., 2015**), not all human studies have found a significant association (**Zhang et al., 2018**; **Stickel et al., 2018**; **Beaudoin et al., 2021**). Importantly, a causal relationship between MBOAT7 and alcohol-induced liver injury has never been established to date. To address this, we have studied ethanol-induced liver injury in mice selectively lacking *Mboat7* in hepatocytes or myeloid cells, given the key roles that hepatocytes and myeloid cells play in the pathogenesis of ethanol-induced liver disease progression. To generate congenic hepatocyte-specific (*Mboat7*-HSKO) and myeloid-specific (*Mboat7*-MSKO) *Mboat7* knockout mice we crossed mice harboring a post-FLP recombinase conditionally targeted *Mboat7* floxed allele (**Anderson et al., 2013**; **Massey et al., 2023**) to mice transgenically expressing Cre recombinase under the albumin promoter/enhancer (**Postic et al., 1999**) or Cre knocked into the M lysozyme locus (**Clausen et al., 1999**), respectively. These independent *Mboat7*-HSKO and *Mboat7*-MSKO lines were then backcrossed mice >10 generations into the C57BL/6J background and subsequently subjected to ethanol exposure. Compared to control mice (*Mboat7*<sup>flox/flox</sup>), *Mboat7*-HSKO mice had significantly reduced *Mboat7* mRNA and protein expression in the liver (**Figure 2A, B**), but not in other tissues (data not shown; **Massey et al., 2023**). Hepatocyte-specific deletion of *Mboat7* resulted in enhanced ethanol-induced increases in liver weight and high concentrations of plasma alanine aminotransferase (ALT) (**Figure 2C, D**). Likewise, *Mboat7*-HSKO mice showed elevated hepatic steatosis scores and triglyceride levels under both pair-fed and ethanol-fed conditions (**Figure 2E, F**). However, hepatocyte-specific deletion of *Mboat7* did not significantly alter the mRNA expression for several proinflammatory cytokines/chemokines including tumor necrosis factor α (*Tnfa*), transforming growth factor β (*Tgfb*), monocyte chemoattractant protein 1 (*Mcp1*), or interleukins 1β (*Il1b*) or 6 (*Il6*) in either pair- or ethanol-fed conditions (**Figure 2G**). These data demonstrate that MBOAT7 function in hepatocytes is a critical determinant of ethanol-induced liver injury, but we also wanted to explore a potential role for MBOAT7 in non-parenchymal cells given the key roles that macrophages and neutrophils play in ALD.

It is important to note during the preparation of this manuscript, an independent study discovered a potential role for MBOAT7 in suppressing toll-like receptor signaling and proinflammatory cytokine production in macrophages and Kupffer cells in the context of NAFLD (**Alharthi et al., 2022**). Furthermore, early studies examining the expression and substrate specificity for diverse lysophospholipid acyltransferases showed that MBOAT7-driven PI remodeling was highly active in human neutrophils where it can modulate the production of proinflammatory arachidonic acid-derived lipid mediators

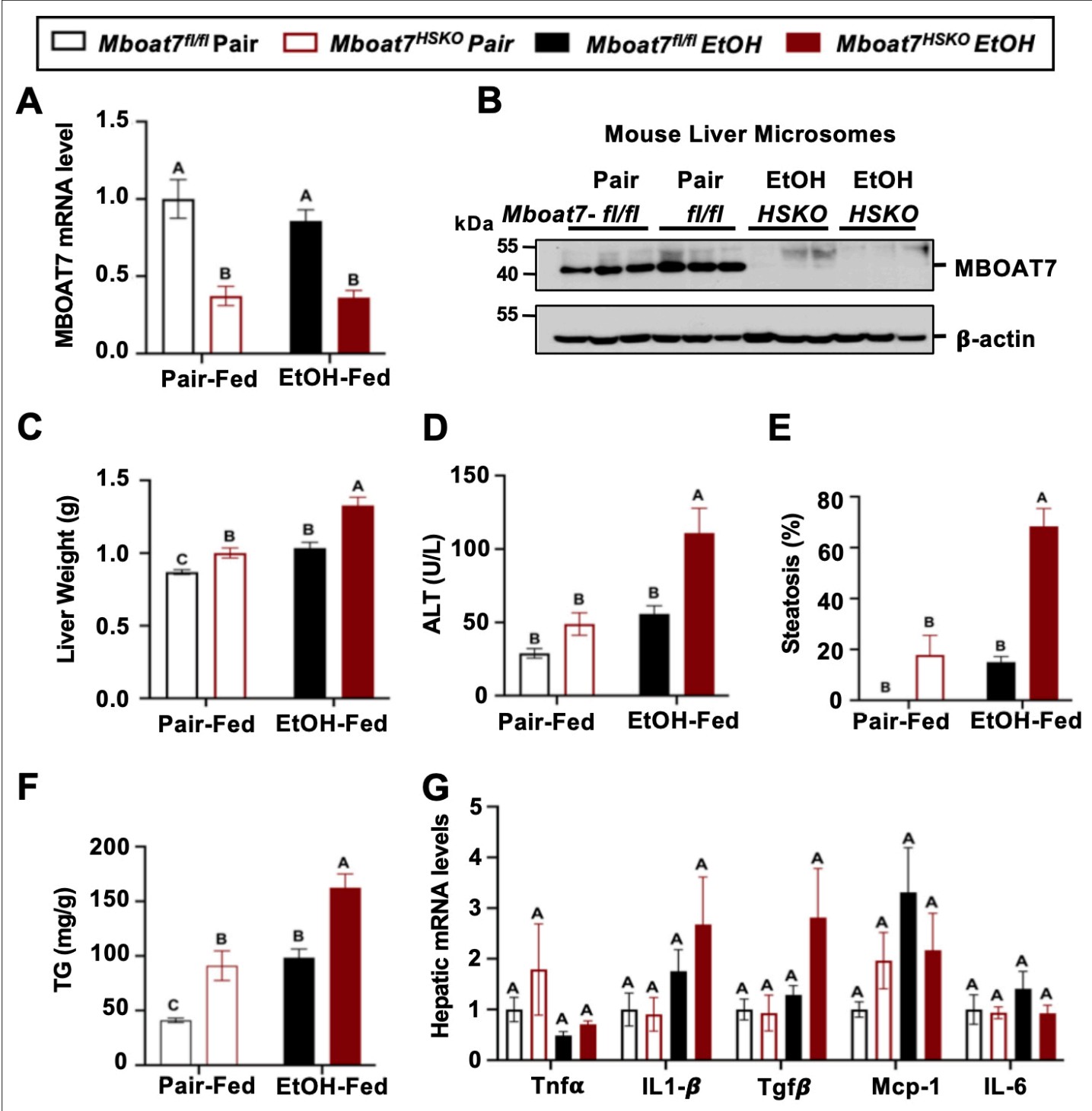

**Figure 2.** Hepatocyte-specific deletion of *Mboat7* promotes ethanol-induced liver injury. Female control (*Mboat7*$^{fl/fl}$) or hepatocyte-specific *Mboat7* knockout mice (*Mboat7*-HSKO) were fed with subjected the NIAAA (National Institute on Alcohol Abuse and Alcoholism) model of ethanol-induced liver injury. (**A**) Hepatic *Mboat7* expression was measured via quantitative polymerase chain reaction (qPCR). (**B**) Western blot for hepatic microsomal MBOAT7 protein levels replicated in *n* = 3 mice. (**C**) Liver weight, (**D**) plasma alanine aminotransferase (ALT), (**E**) percent steatosis quantified by a blinded board-certified pathologist, (**F**) hepatic triglycerides, and (**G**) hepatic expression of inflammatory gene measured by qPCR. *n* = 5–7. Data represent the mean ± standard error of the mean (SEM) and groups not sharing a common letter superscript differ significantly (p ≤ 0.05).

The online version of this article includes the following source data and figure supplement(s) for figure 2:

**Source data 1.** Original file for the western blot analysis in *Figure 2B* (anti-MBOAT7 and anti-β-actin).

*Figure 2 continued on next page*

*Figure 2 continued*

**Source data 2.** PDF containing *Figure 2B* and original scans of the relevant western blot analysis (anti-MBOAT7 and anti-β-actin) with highlighted bands and sample labels.

**Figure supplement 1.** Myeloid-specific deletion of *Mboat7* does not promote ethanol-induced liver injury.

**Figure supplement 1—source data 1.** Original file for the western blot analysis in *Figure 2A* (anti-MBOAT7 and anti-β-actin).

**Figure supplement 1—source data 2.** PDF containing *Figure 2A* and original scans of the relevant western blot analysis (anti-MBOAT7 and anti-β-actin) with highlighted bands and sample labels.

(*Gijón et al., 2008*). Therefore, we generated myeloid-specific (*Mboat7*-MSKO) *Mboat7* knockout mice to further interrogate cell autonomous roles in ALD progression. First to confirm efficient deletion in myeloid cells, we isolated both bone marrow-derived and thioglycolate-elicited macrophages from control and myeloid-specific (*Mboat7*-MSKO) *Mboat7* knockout mice, which confirmed essentially no detectable MBOAT7 protein in *Mboat7*-MSKO mice (*Figure 2—figure supplement 1*). In contrast to the enhanced ethanol-induced liver injury seen in *Mboat7*-HSKO mice (*Figure 2*), myeloid-specific (*Mboat7*-MSKO) *Mboat7* deletion resulted in unaltered ethanol-induced effects on body weight, liver weight, circulating levels of aspartate and alanine aminotransferases (AST and ALT), liver triglyceride, and cytokine expression (*Figure 2—figure supplement 1*). Collectively, these results demonstrate that MBOAT7 loss of function in hepatocytes, but not myeloid cells, facilitates ethanol-induced liver injury in mice.

## Ethanol exposure reorganizes the hepatic lipidome in a MBOAT7-dependent manner

Given the fact that ethanol exposure is well known to reorganize hepatic lipid metabolism, we performed comprehensive lipidomic profiling of the liver to understand how MBOAT7 could potentially shape ethanol-induced lipid metabolism in the liver. First, we used a targeted approach to measure the levels of MBOAT7 substrate LPIs and product PIs. Compared to control mice (*Mboat7*^flox/flox^), *Mboat7*-HSKO mice had significant accumulation of palmitate- and oleate-containing LPI substrate lipids (LPI 16:0 and LPI 18:1), with large accumulation of LPI 16:0 under ethanol-fed conditions (*Figure 3A*). When we examined PI species, we confirmed previous findings that *Mboat7*-HSKO mice have reduced levels of the major arachidonic acid-containing PI (PI 38:4) and also striking reductions in PI 38:3 under both pair- and ethanol-fed conditions (*Figure 3B*). In addition, *Mboat7*-HSKO mice also have accumulation of several other PI species including PI 34:1, PI 36:1, PI 36:2, PI 38:6, and PI 40:6, some of which are exacerbated in the ethanol-fed group (*Figure 3C*). Although many of these changes in MBOAT7's substrate LPIs and product PIs are expected based on MBOAT7's substrate specificity and previous literature, there is a clear interaction between ethanol and MBOAT7 uncovered here that has not been observed in studies using NAFLD-related models (*Tanaka et al., 2021*; *Thangapandi et al., 2021*; *Xia et al., 2021*).

Given this unexpected interaction between ethanol exposure and MBOAT7 within the inositol-containing phospholipid pool, we examined the hepatic lipidome more broadly. Using untargeted lipidomics, we identified a striking remodeling of the global hepatic lipidome upon ethanol feeding in *Mboat7*-HSKO mice (*Figure 3*; *Figure 3—figure supplements 1–11*). Unexpectedly, we observed a large increase in the levels of endosomal/lysosomal lipids including bis(monoacylglycero)phosphates (BMPs) and their outer mitochondrial membrane precursor phosphatidylglycerols (PGs) in ethanol-exposed *Mboat7*-HSKO mice (*Figure 3E–G*; *Figure 3—figure supplements 3 and 4*). In addition, *Mboat7*-HSKO mice had elevated levels of cardiolipin species, which are known to localize to mitochondria (*Figure 3E and H*; *Figure 3—figure supplement 5*). Under the pair- and ethanol-fed conditions studies here, *Mboat7*-HSKO mice also exhibited some more minor alterations in certain species of phosphatidylcholine (PC), phosphatidylethanolamine (PE), phosphatidylserine (PS), phosphatidic acid (PA), sphingomyelin (SM), ceramides (Cer), and ether-linked phospholipids (*Figure 3E*; *Figure 3*, *Figure 3—figure supplements 2; 7–11*). Although several recent studies examining hepatocyte-specific *Mboat7*-HSKO mice have found more limited effects on the global lipidome under experimental conditions designed to stimulate non-alcoholic steatohepatitis (*Tanaka et al., 2021*; *Thangapandi et al., 2021*; *Xia et al., 2021*), here we show that upon ethanol exposure, hepatocyte MBOAT7 plays a major role in shaping endosomal/lysosomal lipid homeostasis.

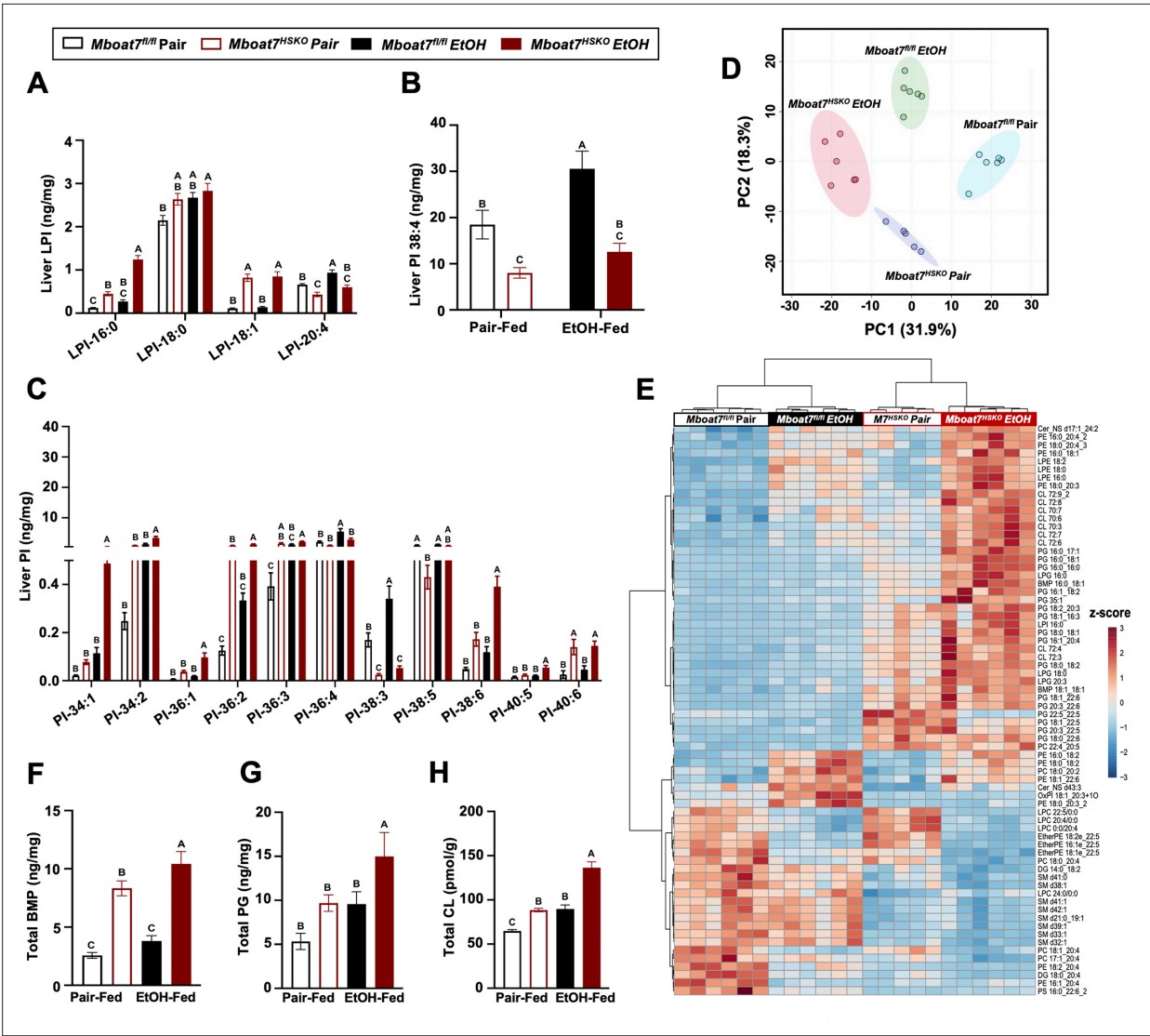

**Figure 3.** Ethanol alters the liver lipidome in a MBOAT7-dependent manner. *Mboat7*[fl/fl] or *Mboat7*-HSKO mice were subjected to the NIAAA model of ethanol exposure. Liver lysophosphatidylinositol (LPI) (**A**) and phosphatidylinositol (PI) species, including the MBOAT7 product PI 38:4 (**B**) and others (**C**), were quantified via liquid chromatography–tandem mass spectrometry (LC–MS/MS) in *n* = 5–7. (**D**) Principal component analysis for untargeted lipidomics analysis. The first and second principal components are plotted on the *x*- and *y*-axis, respectively, and sample treatment group is indicated by color. (**E**) Heatmap showing global lipidomic alterations in mouse liver. Total levels of endosomal/lysosomal lipids were measured by targeted and untargeted lipidomic approach using LC–MS/MS. (**F**) Total bis(monoacylglycero)phosphate (BMP) levels. (**G**) Total phosphatidylglycerol (PG) and (**H**) total cardiolipin (CL) from the liver of Mboat7[fl/fl] or Mboat7-HSKO mice. Data represent the mean ± standard error of the mean (SEM) and groups not sharing a common letter superscript differ significantly (p ≤ 0.05).

The online version of this article includes the following figure supplement(s) for figure 3:

**Figure supplement 1.** Alterations in total hepatic lipid levels in *Mboat7*-HSKO mice.

**Figure supplement 2.** Hepatic phosphatidylcholine (PC) levels in *Mboat7*-HSKO mice.

**Figure supplement 3.** Hepatic bis(monoacylglycerol)phosphate (BMP) levels in *Mboat7*-HSKO mice.

**Figure supplement 4.** Hepatic phosphatidylglycerol (PG) levels in *Mboat7*-HSKO mice.

**Figure supplement 5.** Hepatic cardiolipin (CL) levels in *Mboat7*-HSKO mice.

**Figure supplement 6.** Hepatic phosphatidylserine (PS) levels in *Mboat7*-HSKO mice.

**Figure supplement 7.** Hepatic phosphatidylethanolamine (PE) levels in *Mboat7*-HSKO mice.

**Figure supplement 8.** Hepatic phosphatidic acid (PA) levels in *Mboat7*-HSKO mice.

**Figure supplement 9.** Hepatic sphingomyelin (SM) levels in *Mboat7*-HSKO mice.

*Figure 3 continued on next page*

Figure 3 continued

**Figure supplement 10.** Hepatic ceramide (Cer) levels in *Mboat7*-HSKO mice.

**Figure supplement 11.** Hepatic ether-linked lipids levels in *Mboat7*-HSKO mice.

## *Mboat7*-HSKO mice have dysregulated lysosomal function in response to ethanol

Given the accumulation of endosomal/lysosomal lipids including BMPs seen in ethanol-fed *Mboat7*-HSKO mice, we hypothesized that ethanol may perturb lysosome function in a MBOAT7-driven manner to promote liver injury. It is well known that BMPs commonly accumulate in both drug-induced and genetic lysosomal storage disorders (*Gruenberg, 2020*; *Showalter et al., 2020*; *Hullin-Matsuda et al., 2014*), and due to their cone-shaped structure BMPs can contribute to significant membrane asymmetry that impacts intracellular lipid sorting, apoptosis, and autophagic flux (*Gruenberg, 2020*; *Showalter et al., 2020*; *Hullin-Matsuda et al., 2014*). At the same time, there is emerging evidence that chronic ethanol exposure can reduce the expression of transcription factor EB (TFEB), which is a master regulator of lysosomal biogenesis and autophagy-associated gene expression (*Chao et al., 2018*). Given the role that lysosomal dysfunction plays in ethanol-induced liver injury (*Chao et al., 2018*; *Bala and Szabo, 2018*), and the unexpected accumulation of BMP lipids in *Mboat7*-HSKO mice, we next investigated TFEB-mediated lysosomal biogenesis and autophagy regulation in *Mboat7*-HSKO mice challenged with ethanol (*Figure 4*). First, *Mboat7*-HSKO mice showed elevated levels of key autophagy regulatory proteins LC3-I/II and p62 in the liver, particularly under ethanol-fed conditions (*Figure 4A*). Interestingly, *Mboat7*-HSKO mice also had increased total levels of the mammalian target of rapamycin (mTOR) (*Figure 4A*), which is a well-known master regulator of autophagic flux. The accumulation of p62 and LC3-I/II cannot distinguish between enhanced or defective autophagic flux, so we next examined potential alterations in lysosome abundance and function. Interestingly, both mRNA and protein levels of lysosomal marker proteins LAMP-1 and LAMP-2 were reduced in ethanol-fed *Mboat7*-HSKO mice (*Figure 4A, D*). Furthermore, compared to control mice fed ethanol, we found that ethanol *Mboat7*-HSKO mice had generally reduced expressions levels of TFEB target genes and proteins associated with lysosome acidification and lipid turnover including ATPase H⁺ transporting V1 subunits A, H, and D (*Atp6v1a*, *Atp6v1h*, and *Atp6v1d*), α galactosidase A (*Gla*), chloride channel 7 α (*Clcn7*), and mucolipin TRP cation channel 1 (*Mcoln1*) (*Figure 4A, D*). Similarly, the mRNA expression, and total and nuclear protein abundance of TFEB was reduced in ethanol-fed *Mboat7*-HSKO mice compared to ethanol-fed *Mboat7*^flox/flox^ control mice (*Figure 4A, B and D*). In contrast, the expression of key autophagy related genes including *Atg2b*, *Atg3*, *Atg7*, *Atg8*, and Unc-51-like autophagy activating kinase 1 (*Ulk1*) were significantly elevated in ethanol-fed *Mboat7*-HSKO mice compared to ethanol-fed *Mboat7*^flox/flox^ control mice (*Figure 4D*).

We next investigated the cell autonomous effects of ethanol on wild-type human Huh7 hepatoma cells or Huh7 cells genetically lacking *MBOAT7* (Huh7-Δ *MBOAT7*). In agreement with what we found in mouse liver, *MBOAT7*Δ-Huh7 cells had reduced levels of total TFEB and lysosomal marker proteins including LAMP1-, LAMP-2, and ATP6V1A (*Figure 4—figure supplement 1*). Furthermore, *MBOAT7*Δ-Huh7 cells had increased levels of LC3 and total mTOR, which was particularly apparent upon ethanol exposure (*Figure 4—figure supplement 1*). We next assessed lysosome protein degradation activity in wild-type and MBOAT7Δ-Huh7 hepatoma cells by measuring the degradation of an exogenous lysosome indicator, quantified via median Bodipy/Alexa647 fluorescence ratio. In the absence of ethanol treatment, there was no significant difference in the lysosomal activity between wild-type and MBOAT7Δ-Huh7 cells. Lysosome activity was increased by 42% in wild-type cells with ethanol treatment vs. no treatment (p < 0.001, *Figure 4C*), in agreement with increased mRNA levels of several lysosomal hydrolases (*Figure 4D*). However, the lysosome activity was decreased by 45% in MBOAT7Δ-Huh7 vs. wild-type cells treated with ethanol (p < 0.001, *Figure 4C*). Collectively, in the presence of EtOH, deletion of *MBOAT7* in mouse or human hepatocytes results in defective TFEB-mediated lysosomal biogenesis and lysosome activity, which would be expected to lead to impaired autophagic flux.

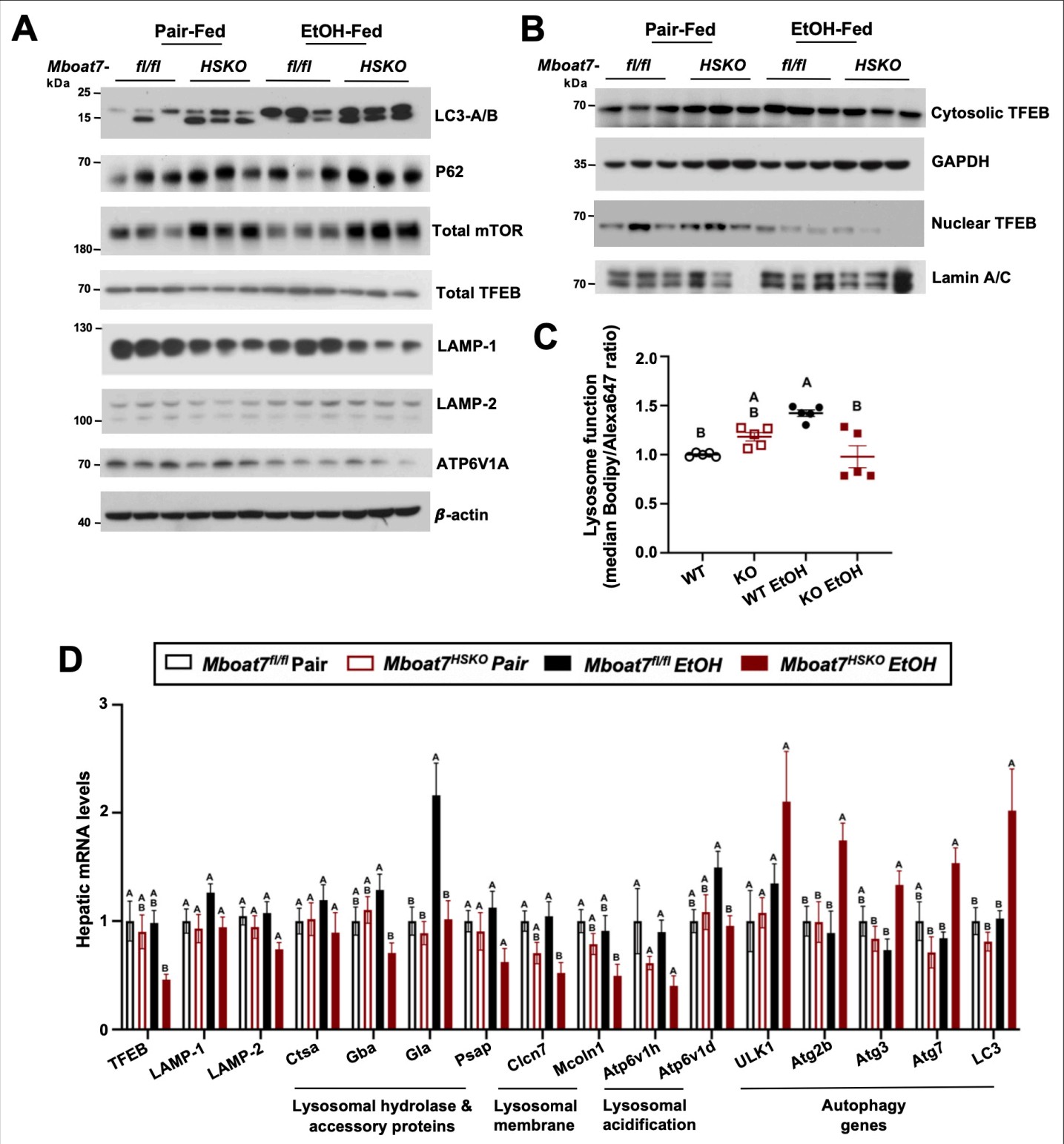

**Figure 4.** *Mboat7*-HSKO mice have dysregulated lysosome function in response to ethanol. Age-matched female *Mboat7*<sup>fl/fl</sup> or *Mboat7*-HSKO mice were subjected to the NIAAA model of ethanol exposure. (**A**) Total liver lysates were subjected to western blot analysis of major autophagy marker genes LC3A/B (P62), mammalian target of rapamycin (mTOR) and lysosome biogenesis genes (TFEB, LAMP-1, LAMP-2, and ATP6V1A). (**B**) Nuclear fractions from mouse livers of Mboat7<sup>fl/fl</sup> and *Mboat7*-HSKO were subjected to western blot analysis of TFEB. (**C**) Lysosome protein degradation activity in wild-type and MBOAT7Δ-Huh7 hepatoma cells treated with or without 100 mM ethanol for 48 hr was assessed by incubating cells with 10 µg/ml of lysosome indicator for 2 hr and examined by flow cytometry. *n* = 5 from two experiments by normalizing to wild-type group in each experiment; mean ±

*Figure 4 continued on next page*

*Figure 4 continued*

standard deviation (SD) (**D**) Expression levels of the genes encoding functions in lysosomal hydrolase and accessory, lysosomal m involved in lysosomal biogenesis in the liver of Mboat7*fl/fl* and Mboat7-HSKO *mice* upon ethanol feeding. mRNA expression levels were determined by qPCR (*n* = 6/group). Groups not sharing a common letter superscript differ significantly (p ≤ 0.05).

The online version of this article includes the following source data and figure supplement(s) for figure 4:

**Source data 1.** Original file for the western blot analysis in *Figure 4A* (anti-LC3-A/B).

**Source data 2.** Original file for the western blot analysis in *Figure 4A* (anti-p62).

**Source data 3.** Original file for the western blot analysis in *Figure 4A* (anti-Total mTOR and anti-LAMP-1).

**Source data 4.** Original file for the western blot analysis in *Figure 4A* (anti-TFEB).

**Source data 5.** Original file for the western blot analysis in *Figure 4A* (anti-LAMP-2 and anti-ATP6V1A).

**Source data 6.** Original file for the western blot analysis in *Figure 4A* (anti-β-actin).

**Source data 7.** PDF containing *Figure 4A* and original scans of the relevant western blot analysis (anti-MBOAT7, anti-p62, anti-total mTOR, anti-total TFEB, anti-LAMP-1, anti-LAMP-2, anti-ATP6V1A, and anti-β-actin) with highlighted bands and sample labels.

**Source data 8.** Original file for the western blot analysis in *Figure 4B* (anti-cytosolic-TFEB).

**Source data 9.** Original file for the western blot analysis in *Figure 4B* (anti-nuclear TFEB).

**Source data 10.** Original file for the western blot analysis in *Figure 4B* (anti-LAMIN A/C).

**Source data 11.** Original file for the western blot analysis in *Figure 4B* (anti-GAPDH).

**Source data 12.** PDF containing *Figure 4B* and original scans of the relevant western blot analysis (anti-cytosolic TFEB, anti-nuclear TFEB, anti-LAMIN A/C, and anti-GAPDH) with highlighted bands and sample labels.

**Figure supplement 1.** Genetic deletion of MBOAT7 in human Huh7 cells is associated with diminished lysosome biogenesis and ethanol-induced autophagy dysregulation.

**Figure supplement 1—source data 1.** Original file for the western blot analysis in *Figure 4A, B* (anti-LC3-A/B).

**Figure supplement 1—source data 2.** Original file for the western blot analysis in *Figure 4A and B* (anti-p62).

**Figure supplement 1—source data 3.** Original file for the western blot analysis in *Figure 4A and B* (anti-Total mTOR and anti-LAMP-1).

**Figure supplement 1—source data 4.** Original file for the western blot analysis in *Figure 4A and B* (anti-TFEB).

**Figure supplement 1—source data 5.** Original file for the western blot analysis in *Figure 4A and B* (anti-LAMP-2 and anti-ATP6V1A).

**Figure supplement 1—source data 6.** Original file for the western blot analysis in *Figure 4A and B* (anti-β-actin).

**Figure supplement 1—source data 7.** PDF containing *Figure 4A and B* and original scans of the relevant western blot analysis (anti-MBOAT7, anti-p62, anti-total mTOR, anti-total TFEB, anti-LAMP-1, anti-LAMP-2, anti-ATP6V1A, and anti-β-actin) with highlighted bands and sample labels.

**Figure supplement 2.** Working model.

## Discussion

This manuscript builds on our initial observation that ASO-mediated knockdown of *Mboat7* promotes NAFLD progression, hyperinsulinemia, and insulin resistance in mice (*Gwag et al., 2019*). Here, we have further clarified the cell autonomous roles of *Mboat7* in ethanol-driven liver injury by comparing metabolic phenotypes in hepatocyte-specific (*Mboat7*-HSKO) and myeloid-specific (*Mboat7*-MSKO) mice. The major findings of the current study include the following: (1) MBOAT7 product PI species (PI 38:4 and PI 38:5) are reduced in the circulation of human consuming high levels of alcohol; (2) MBOAT7 loss of function in hepatocytes, but not myeloid cells, promotes ethanol-induced liver injury in mice; (3) hepatocyte-specific deletion of *Mboat7* results in expected alterations in substrate LPI and product PI lipids, but unexpectedly alters lysosomal/endosome BMP lipids in an ethanol-driven manner; (4) genetic deletion in mouse or human hepatocytes results in dysregulation of lysosomal biogenesis and autophagic flux, particularly under ethanol-challenged conditions. This work provides new insights into how genetic variation in *MBOAT7* may impact ALD progression in humans and mice. Importantly, this work is the first to causally link MBOAT7 loss of function in hepatocytes, but not myeloid cells, to ethanol-induced liver injury via dysregulation of lysosomal biogenesis and autophagic flux. Although not all human studies agree there is a uniform association between the rs641738 SNP with ALD, our work indicates a very powerful interaction between MBOAT7 loss of function and ethanol-induced liver injury.

Here, we have identified a striking reorganization of the hepatic lipidome in *Mboat7*-HSKO mice when exposed to ethanol (*Figure 3*; *Figure 3—figure supplements 1–11*). In previous independent

studies examining lipid alterations in *Mboat7*-HSKO mice under experimental conditions to elicit NAFLD and fibrosis (i.e., high fat diets or methionine/choline-deficient diets), the global lipidomic alterations in the liver were much more confined to inositol-containing phospholipids and triacylglycerols (*Tanaka et al., 2021*; *Thangapandi et al., 2021*). It is important to note that the work of Xia and colleagues did previously report increased levels on PG lipids specifically in isolated endoplasmic reticulum (ER) membranes from *Mboat7*-HSKO mice, but our work confirms and extends this to show that not only PG but precursor BMPs are significantly elevated in *Mboat7*-HSKO mice, particularly when challenged with ethanol. It is still unclear how MBOAT7 impacts endosomal/lysosomal BMP and mitochondrial lipids such as PG and CL under ethanol-exposed conditions, but our work clearly indicates that ethanol reorganizes the global liver lipidome in a MBOAT7-dependent manner. It is most likely that the accumulation of BMP and PG lipids seen in ethanol-challenged *Mboat7*-HSKO mice are not directly related to the lysophospholipid acyltransferase activity of MBOAT7. Instead, it is more plausible that the accumulation of BMPs and PGs seen in *Mboat7*-HSKO mice is secondary to indirect reorganization of the arachidonate PI cycle or other related lipid metabolic pathways coordinated at the ER.

For instance, the arachidonate PI cycle is initiated in the ER where inositol is added to (cytidine diphosphate)CDP-diacylglycerol (18:0/20:4) by phosphatidylinositol synthase (PIS) to produce the exact same metabolic product of MBOAT7 (PI 38:4). It is likely that both PIS-generated as well as MBOAT7-generated PI 38:4 can serve as a substrate for PI kinases to form the key second messengers known as PI phosphates [PIPs including PI(18:0/20:4)-4P, PI(18:0/20:4)-4,5$_{P2}$] and related lipid mediators downstream of phospholipase C in the arachidonate PI cycle IP3, DAG(18:0/20:4), PA(18:0/20:4), and CDP-DAG(18:0/20:4). It is important to note that seminal work by *Anderson et al., 2013* found that total PIPs, PI(18:0/20:4)-4P, and PI(18:0/20:4)-4,5$_{P2}$ were significantly reduced in global *Mboat7*$^{-/-}$ mice. In fact, more than 85% of PIP species in cultured cells have an *sn*-1 18:0 and *sn*-2 20:4 acyl chain composition (i.e., in part originate from the MBOAT7 and PIS product PI 38:4) (*Clark et al., 2011*; *Rouzer et al., 2007*). These reductions in PIPs seen with *Mboat7* deficiency (*Anderson et al., 2013*) could have important consequences in cellular signal transduction, given that PIPs are common second messengers generated downstream of ligand activation of numerous receptor systems including hormone, growth factor, cytokine, and chemokine receptors (*Wymann and Schneiter, 2008*; *Pemberton et al., 2020*; *Fruman et al., 2017*; *Hoxhaj and Manning, 2020*). PIPs also play diverse roles in shaping protein–lipid interactions, membrane fusion events, vesicular transport, solute channel function, and cytoskeletal arrangement (*Wymann and Schneiter, 2008*; *Pemberton et al., 2020*; *Fruman et al., 2017*; *Hoxhaj and Manning, 2020*). Most relevant to this work, anionic lipids play very important roles in controlling membrane dynamics that shape nearly all steps of autophagy including initiation of autophagosome biogenesis and autophagosome–lysosome fusion (*Baba and Balla, 2020*; *Schink et al., 2016*). Collectively, given the fact that MBOAT7 generates the most abundant species of PI (PI 38:4), and key cellular PIPs [PI(18:0/20:4)-4P and PI(18:0/20:4)-4,5$_{P2}$], there is a strong potential that the primary alterations in PI and PIP lipids could broadly alter cellular signal transduction, endosomal/lysosomal lipid sorting, membrane fusion events, vesicular transport, solute channel function, cytoskeletal arrangement, and autophagic flux.

Collectively, this work provides new cellular and molecular insights into how genetic variation in *MBOAT7* impacts ALD progression in humans and mice. This work is the first to causally link MBOAT7 loss of function in hepatocytes, but not myeloid cells, to ethanol-induced liver injury via dysregulation of lysosomal biogenesis and autophagic flux and broaden our understanding of the lipid metabolic mechanisms promoting ethanol-induced liver injury. This work also shows that MBOAT7-driven LPI acylation in the ER can indirectly impact both lysosomal (BMP) and mitochondrial (CL and PG) lipids which can have broad impacts on autophagy described here, as well as defective fatty acid oxidation as we originally reported in high fat diet-fed mice (*Gwag et al., 2019*). The results of this work have broad potential implications in the management of both alcoholic- and non-alcoholic fatty liver disease, indicating that strategies that effectively restore both lysosomal and mitochondrial function may hold some therapeutic promise in humans with the common MBOAT7 rs641738 variant.

## Materials and methods

**Key resources table**

| Reagent type (species) or resource | Designation | Source or reference | Identifiers | Additional information |
|---|---|---|---|---|
| Genetic reagent (*M. musculus*) | *Mboat7tm1a(KOMP)Wtsi/ Mboat7tm1a(KOMP)Wtsi* | PMID:23472195 | RRID: MGI:5510874 | |
| Genetic reagent (*M. musculus*) | *B6N.Cg-Speer6-ps1*<sup>*Tg(Alb-cre)21Mgn*</sup>*/J* | Jackson Laboratory | Stock#: 018961 RRID: IMSR_ JAX:018961 | |
| Genetic reagent (*M. musculus*) | *B6.129P2-Lyz2*<sup>*tm1(cre)Ifo*</sup>*/J* | Jackson Laboratory | Stock#: 004781 RRID: IMSR_ JAX:004781 | |
| Cell line (*Homo sapiens*) | HUH7 (well differentiated human hepatocellular carcinoma) | Japanese Collection of Research Biosources Cell Bank | JCRB0403 | |
| Antibody | Anti-MBOAT7 (Rat monoclonal) | PMID:23097495 | RRID: AB_2813851 | WB (1:1000) |
| Antibody | Anti-rat IgG HRP secondary antibody | Cell Signaling | Cat#: 7077 RRID: AB_10694715 | WB (1:5000) |
| Antibody | LC3A/B (D3U4C) XP (Rabbit monoclonal) | Cell Signaling | Cat#: 12741 RRID: AB_2617131 | WB (1:1000) |
| Antibody | SQSTM1/p62 (Rabbit polyclonal) | Cell Signaling | Cat#: 5114 RRID: AB_10624872 | WB (1:1000) |
| Antibody | mTOR (7C10) (Rabbit monoclonal) | Cell Signaling | Cat#: 2983 RRID: AB_2105622 | WB (1:1000) |
| Antibody | TFEB (D2O7D) (Rabbit monoclonal) | Cell Signaling | Cat#: 37785 | WB (1:1000) |
| Antibody | TFEB (Rabbit polyclonal) | Thermo Fisher Scientific | Cat#: A303-673A RRID: AB_11204751 | WB (1:1000) |
| Antibody | LAMP1 (D2D11) XP Rabbit monoclonal | Cell Signaling | Cat#: 9091 RRID: AB_2687579 | WB (1:1000) |
| Antibody | LAMP-1 (Rat monoclonal) | Developmental Studies Hybridoma Bank (DSHB) | Cat#: ID4B RRID: AB_528127 | WB (1:1000) |
| Antibody | LAMP-2 (Rat monoclonal) | Developmental Studies Hybridoma Bank (DSHB) | Cat#: ABL-93 RRID: AB_2134767 | WB (1:1000) |
| Antibody | LAMP2 (D5C2P) (Rabbit monoclonal) | Cell Signaling | Cat#: 49067 RRID: AB_2799349 | WB (1:1000) |
| Antibody | ATP6V1A (Rabbit polyclonal) | GeneTex | Cat#: GTX110815 RRID: AB_1949704 | WB (1:1000) |
| antibody | Anti-GAPDH-HRP (Rabbit monoclonal) | Cell Signaling | Cat#: 8884 RRID: AB_11129865 | WB (1:5000) |
| Antibody | Lamin A/C (4C11) (Mouse monoclonal) | Cell Signaling | Cat#: 4777 | WB (1:1000) |

| Reagent type (species) or resource | Designation | Source or reference | Identifiers | Additional information |
|---|---|---|---|---|
| Antibody | HRP-conjugated Beta Actin (Mouse monoclonal) | Proteintech | Cat#: HRP-60008 RRID: AB_2289225 | WB (1:10,000) |
| Commercial assay or kit | Alanine Aminotransaminase (ALT) kit | Sekisui Diagnostics | 318-30 | |
| Commercial assay or kit | Aspartate Aminotransferase (AST) kit | Sekisui Diagnostics | 319-30 | |
| Commercial assay or kit | Liver Triglyceride | Wako | 994-02891 | |
| Commercial assay or kit | Microsome Isolation | Abcam | ab206995 | |
| Commercial assay or kit | NE-PER Nuclear and Cytoplasmic Extraction Reagents | Thermo Fisher Scientific | 78833 | |
| Commercial assay or kit | Supersignal West Pico Plus substrate | Thermo Fisher Scientific | 34577 | |
| Chemical compound, drug | Ammonium formate | Honeywell | Cat# 55674 | |
| Chemical compound, drug | Methanol | Honeywell | Cat# LC230-4 | |
| Chemical compound, drug | Water | Honeywell | Cat# LC365-4 | |
| Chemical compound, drug | Acetonitrile | Honeywell | Cat# LC015-4 | |
| Chemical compound, drug | Isopropanol | Fisher Scientific | Cat# A461-4 | |
| Chemical compound, drug | Ethyl acetate | Sigma-Aldrich | Cat# 650528 | |
| Chemical compound, drug | Formic acid | Thermo Scientific | Cat# 28905 | |
| Chemical compound, drug | FA 18:0$_{d35}$ | Cayman Chemical | Cat# 9003318 | |
| Chemical compound, drug | ACar 18:1$_{d3}$ | Cayman Chemical | Cat# 26578 | |
| Chemical compound, drug | BMP 14:0_14:0 | Avanti | Cat# 857131 | |
| Chemical compound, drug | PG 15:0_18:1$_{d7}$ | Avanti | Cat# 91640 | |
| Chemical compound, drug | Cer d18:1$_{d7}$_15:0 | Avanti | Cat# 860681P | |
| Chemical compound, drug | PA 15:0_18:1$_{d7}$ | Avanti | Cat# 791642 | |
| Chemical compound, drug | SPLASH LipidoMix II | Avanti | Cat# 330709 | |
| Chemical compound, drug | BMP 18:1_18:1 | Avanti | Cat# 857133P | |
| Chemical compound, drug | PG 18:1_18:1 | Avanti | Cat# 840475P | |

## Human studies

*Healthy Control and Heavy Drinking Patient Selection* – Healthy controls or heavy drinkers with an AUDIT score greater than >16 were recruited from the Clinical Research Unit at the Cleveland Clinic or MetroHealth Hospital in Cleveland, Ohio based on medical history and physical examination. The study protocol was approved by the Institutional Review Board for the Protection of Human Subjects in Research at the Cleveland Clinic (IRB 17-718) and MetroHealth Hospital Cleveland (IRB 18-00911). All methods were performed in accordance with the internal review board's guidelines and regulations, and written, informed consent was obtained from all subjects. Subject demographics are shown in *Figure 1—source data 1*.

## Mice and experimental diets

To generate conditional *Mboat7* knockout mice, we obtained 'knockout first' (Mboat7$^{tm1a(KOMP)Wtsi}$) mice from Dr. Philip Hawkins (*Anderson et al., 2013*), and crossed these mice with mice transgenically expressing FLP recombinase to remove the NEO cassette resulting in a conditional *Mboat7* floxed allele. The FLP transgene was then subsequently bred out of the line and resulting *Mboat7*$^{flox/WT}$ mice, which were used to expand further downstream tissue-specific knockout lines. To generate congenic hepatocyte-specific (*Mboat7*-HSKO) and myeloid-specific (*Mboat7*-MSKO) *Mboat7* knockout mice we crossed mice harboring a post-FLP recombinase conditionally targeted *Mboat7* floxed allele (*Anderson et al., 2013*; *Massey et al., 2023*) to mice transgenically expressing Cre recombinase under the albumin promoter/enhancer (*Postic et al., 1999*) or Cre knocked into the M lysozyme locus (*Clausen et al., 1999*), respectively. These independent *Mboat7*-HSKO and *Mboat7*-MSKO lines were then backcrossed mice >10 generations into the C57BL/6J background and subsequently subjected to ethanol exposure. Confirmation of sufficient backcrossing into the C57BL/6J background was confirmed by mouse genome SNP scanning at the Jackson Laboratory (Bar Harbor, ME). Age- and weight-matched female (8–10 weeks old) control (Mboat7$^{flox/flox}$), hepatocyte-specific *Mboat7* knockout mice (Mboat7-HSKO), or myeloid-specific *Mboat7* knockout mice (Mboat7-MSKO) were maintained on a chow diet and randomized into pair- and ethanol-fed groups using the NIAAA model (*Bertola et al., 2013*). Briefly, mice were initially fed with control Lieber–DeCarli diet ad libitum for 5 days to acclimatize them to liquid diet. Afterward, ethanol (EtOH)-fed groups were allowed free access to the ethanol Lieber–DeCarli diet containing 5% (vol/vol) ethanol for 10 days, and control groups were pair-fed with the isocaloric substituted maltose dextrins as control diet. At day 11, ethanol- and pair-fed mice were gavaged in the early morning with a single dose of ethanol (5 g/kg body weight) or isocaloric maltose dextrin, respectively, and euthanized 6 hr later (*Bertola et al., 2013*). All mice were maintained in an Association for the Assessment and Accreditation of Laboratory Animal Care, International-approved animal facility, and all experimental protocols were approved by the Institutional Animal Care and Use Committee of the Cleveland Clinic (IACUC protocols # 2018-2053 and # 00002499).

## Histological analysis and imaging

Hematoxylin and eosin (H&E) staining of paraffin-embedded liver sections was performed as previously described (*Gromovsky et al., 2017*; *Warrier et al., 2015*; *Brown et al., 2010*; *Brown et al., 2008a*; *Brown et al., 2008b*). Histopathologic evaluation was scored in a blinded fashion by a board-certified pathologist with expertise in gastrointestinal/liver pathology (Daniela S. Allende – Cleveland Clinic). H&E slides were scanned using a Leica Aperio AT2 Slide Scanner (Leica Microsystems, GmbH, Wetzlar, Germany) and images were processed using ImageScope (Aperio, Software Version 12.1).

## Immunoblotting

Whole tissue homogenates were made from tissues in a modified RIPA buffer as previously described (*Warrier et al., 2015*; *Gwag et al., 2019*; *Schugar et al., 2017*; *Lord et al., 2016*), microsome was isolated from the livers of control (*Mboat7*$^{fl/fl}$) or hepatocyte-specific *Mboat7* knockout mice (*Mboat7*-HSKO) using microsome isolation kit from Abcam and protein was quantified using the bicinchoninic assay (Pierce). Proteins were separated by 4–12% sodium dodecyl sulfate–polyacrylamide gel electrophoresis transferred to polyvinylidene difluoride membranes, and then proteins were detected after incubation with specific antibodies as previously described (*Warrier et al., 2015*; *Gwag et al., 2019*; *Schugar et al., 2017*; *Lord et al., 2016*) and listed in the Key resources table.

## Real-time PCR analysis of gene expression

Tissue RNA extraction and qPCR analysis were performed as previously described (*Gwag et al., 2019*). The mRNA expression levels were calculated based on the ΔΔCT method using cyclophilin A as the housekeeping gene. qPCR was conducted using the Applied Biosystems 7500 Real-Time PCRsystem. All primer sequences can be found in *Supplementary file 1*.

## Plasma and liver biochemistries

To determine the level of hepatic injury in mice fed HFD, plasma was used to analyze ALT levels using enzymatic assays as previously described (*Gwag et al., 2019*). Extraction of liver lipids and quantification of total plasma and hepatic triglycerides were conducted using enzymatic assays as described previously (*Gwag et al., 2019*; *Gromovsky et al., 2017*).

## Liver lipid extraction

Lipids were extracted from liver samples as previously described (*Jain et al., 2022*). Samples were kept on ice throughout the extraction. Briefly, liver was homogenized in 500 µl of 3:1:6 isopropanol:water:ethyl acetate containing internal standard in ceramic bead tubes (QIAGEN #13113-50) using the TissueLyzer II (QIAGEN #9244420). Samples were centrifuged at 16,000 × *g* for 10 min at 4°C and the lipid containing supernatant was transferred to a new 1.5 ml tube. Lipid extracts were dried in a SpeedVac (Thermo Savant RVT5105) and resuspended in 150 µl of methanol. Samples were kept at 4°C for no more than 1 week before analysis.

## Targeted quantification of LPI and PI lipids

Quantitation of LPI and PI species was performed as previously described (*Gwag et al., 2019*) Briefly, LPI and PI standards (LPI 16:0, LPI 18:0, LPI 18:1, LPI 20:4, PI 38:4) and the two internal standards (LPI 17:1-d31, PI 34:1-d31) were purchased Avanti Polar Lipids. High-performance liquid chromatography (HPLC) grade water, methanol, and acetonitrile were purchased from Thermo Fisher Scientific. Standard LPI and PI species at concentrations of 0, 5, 20, 100, 500, and 2000 ng/ml were prepared in 90% methanol containing two internal standards at the concentration of 500 ng/ml. Samples were injected into the Shimadzu LCMS-8050 for generating the internal standard calibration curves. A triple quadrupole mass spectrometer (Quantiva, Thermos Fisher Scientific, Waltham, MA, USA) was used for analysis of LPI and PI species. The mass spectrometer was coupled to the outlet of an UHPLC system (Vanquish, Thermos Fisher Scientific, Waltham, MA, USA), including an auto sampler with refrigerated sample compartment and inline vacuum degasser. The HPLC eluent was directly injected into the triple quadrupole mass spectrometer and the analytes were ionized at ESI(Electrospray Ionization) negative mode. Analytes were quantified using Selected Reaction Monitoring (SRM) and the SRM transitions (*m/z*) were 571 → 255 for LPI 16:0, 599 → 283 for LPI 18:0, 597 → 281 for LPI 18:1, 619 → 303 for LPI 20:4, 885 → 241 for PI 38:4, 583 → 267 for internal standard LPI 17:1, and 866 → 281 for internal standard PI 34:1-d31. Xcalibur software was used to get the peak area for both the internal standards and LPI and PI species. The internal standard calibration curves were used to calculate the concentration of LPI and PI species in the samples.

## Untargeted lipidomics

Untargeted lipidomics was performed using an Agilent 1290 Infinity II liquid chromatograph equipped with a Waters Acquity BEH C18 column (1.7 µm 2.1 × 100 mm) coupled to an Agilent 6546 Q-TOF mass spectrometer. Mobile phase A was 60:40 acetonitrile:water and B was 90:9:1 isopropanol:acetonitrile:water with both phases buffered with 10 mM ammonium formate and 0.1% formic acid. The gradient was as follows: starting at 15% B to 30% B at 2.40 min, 48% at 3 min, 82% at 13.2 min, 99% at 13.8 min then held at 99% until 15.4 min before equilibrating 15%, held until 20 min. Samples were analyzed in both positive and negative ionizations in separate experiments with the following MS parameters: drying gas flowing 12 l/min at 250°C, nebulizer at 30 psi and sheath gas flowing 11 l/min at 300°C for positive mode. The sheath gas flow was 12 l/min at 375°C with a nebulizer pressure of 30 psi in negative mode. Both ionizations had the same voltage for capillary (4000 V), skimmer (75 V), fragmentor (190 V), and octopole (750 V). Reference masses (*m/z* = 121.05 and 922 for positive mode; 112.98 and 966.00 for negative mode) were continuously infused during sample runs for accurate mass calibration. Pooled samples were run in consecutive iterative MS/MS injections at a constant

collision energy of 25 V to collect spectra for lipid library creation using Agilent LipidAnnotator and MS1 data were collected for all individual samples. Peak integration was performed using Agilent Profinder (v8.0) software and further curation and internal standard normalization were performed using in-house R scripts as described previously (*Jain et al., 2022*). Untargeted data are reported in units of pmol lipid/g tissue. Principal component analysis and heatmaps were generated using MetaboAnalyst (*Pang et al., 2021*). The first and second principal components are plotted on the *x*- and *y*-axis, respectively, and sample treatment group is indicated by color. For heatmap generation, data were pareto-scaled and the normalized intensity is indicated by color with relative increase in red and decrease in blue. The top 70 lipid features by analysis of variance (ANOVA) p-value are plotted on the *y*-axis and samples are grouped by Ward clustering on the *x*-axis. Sample treatments are distinguished by color.

## Targeted quantification of BMPs and PGs

Targeted BMP and PG lipid analysis was conducted on an Agilent 1290 Infinity II LC coupled to an Agilent 6495C QQQ mass spectrometer. The same column, mobile phases, and gradient were used for targeted analysis as for the untargeted. The MS parameters for both ionizations were as follows: drying gas temp flow of 12 l/min at 250°C with nebulizer at 35 psi and sheath gas flow of 11 l/min at 300°C. Capillary voltage was kept at 4000 V with nozzle voltage of 500 V and collision energy = 20 V. The iFunnel high pressure RF was at 150/90 V and low-pressure RF at 60/60 V for positive/negative ionizations, respectively. A sample type specific lipid library was created by running pooled liver extract multiple times in positive ionization using multiple reaction monitoring (MRM) to scan for a list of BMP/PG lipid precursor masses ($[NH4]^+$ adducts) with transitions for the respective fatty acid and diacylglycerol ($-17$ *m/z*) adducts (*Grabner et al., 2019*). BMPs were identified by the characteristic free fatty acid fragment being much higher intensity than the diacylglycerol fragment, whereas the opposite was true for PG. In addition, BMP lipids eluted between 0.3 and 0.5 min earlier than the isomeric PG counterpart. These distinguishing characteristics were validated using commercial BMP 18:1_18:1 and PG 18:1_18:1 standards. A dynamic MRM method was created using the retention time data gathered from positive ionization tests but in negative mode scanning for the $[M-H]^-$ precursors and fatty acid fragments as BMP and PG are much more readily ionized in negative mode. Data were integrated in Agilent Quantitative analysis and peaks were manually adjusted and verified. Final units are in ng lipid/mg tissue.

## Lysosome protein degradation activity assay

Lysosome protein degradation activity was performed as previously described (*Robinet et al., 2021*). Briefly, DQ-ovalbumin (D12053, Thermo Fisher Scientific) was labeled with Alexa Fluor 647 succinimidyl ester (A20006, Thermo Fisher Scientific) at room temperature for 1 hr (3:1 dye:protein mole ratio) to make lysosome protein degradation indicator. The reaction was stopped by incubating the conjugate with 1.5 M hydroxylamine (pH 8.5) for 1 hr at room temperature, and the conjugate was purified by extensive dialysis. Lysosome protein degradation indicator was validated in vitro by incubating with proteinase K and achieving significant increase of Bodipy/Alexa647 ratio (*Robinet et al., 2021*). The double labeled ovalbumin has increased Bodipy fluorescence upon its degradation by decreasing its self-quenching, while Alexa647 is not changed upon its degradation and is used to normalize for cellular uptake. Wild-type or MBOAT7Δ-Huh7 hepatoma cells treated with or without 100 mM ethanol for 48 hr were incubated with 10 µg/ml of lysosome indicator for 2 hr, and lysosome activity was analyzed by flow cytometry in ≈10,000 cells with a LSRFortessa device (BD). Flowjo software was used to export data for each cell for ratiometric analyses, and the median Bodipy/Alexa647 ratio for each independent well was used for analysis.

## Statistical analyses

Single comparisons between two groups were performed using two-tailed Student's *t* tests with 95% confidence intervals. Comparisons involving multiple groups beyond binary comparison were assessed using one-way ANOVA with Tukey's post hoc test. All data presented as mean ± standard error of the mean. Values were considered significant at $p < 0.05$ (using superscripts), or ***$p < 0.001$ and ****$p < 0.0001$ in *Figure 1*. JMP 17.0 statistical discovery software (SAS Institute, Cary, NC, USA) was used for all statistical analyses.

## Acknowledgements

This work was supported in part by National Institutes of Health grants P50 AA024333 (J.D., S.D., D.S.A., L.E.N., J.M.B.), R01 DK120679 (J.M.B.), RF1 NS133812 (J.M.B.), P01 HL147823 (J.M.B.), U01 AA026938 (L.E.N., J.M.B.), R01 DK130227 (J.M.B.), U01 AA026264 (L.E.N.), R01 HL128268 (J.D.S.), JDRF JDRF201309442 (J.S.), the Glenn and AFAR Junior Faculty Grant A22068 (J.S.), R01 DK133479 (J.S.); Hatch Grant WIS04000 (R.J., J.S.), K01 DK128022 (R.N.H.), UL1TR001998 (R.N.H.), and the American Heart Association-Postdoctoral Fellowships 19POST34380725 (I.R.). The authors would like to thank Dr. Philip Hawkins (Babraham Institute) for providing *Mboat7* knockout first mice.

## Additional information

### Competing interests

Judith Simcox, J Mark Brown: Reviewing editor, *eLife*. The other authors declare that no competing interests exist.

### Funding

| Funder | Grant reference number | Author |
| --- | --- | --- |
| NIH Office of the Director | P50 AA024333 | Srinivasan Dasarathy<br>Daniela S Allende<br>Jonathan D Smith<br>Laura E Nagy<br>J Mark Brown |
| NIH Office of the Director | R01 DK120679 | J Mark Brown |
| NIH Office of the Director | RF1 NS133812 | J Mark Brown |
| NIH Office of the Director | P01 HL147823 | J Mark Brown |
| NIH Office of the Director | U01 AA026938 | Laura E Nagy<br>J Mark Brown |
| NIH Office of the Director | R01 DK130227 | J Mark Brown |
| NIH Office of the Director | U01 AA026264 | J Mark Brown |
| NIH Office of the Director | R01 HL128268 | Jonathan D Smith |
| JDRF | JDRF201309442 | Judith Simcox |
| Glenn Foundation for Medical Research | A22068 | Judith Simcox |
| NIH Office of the Director | R01 DK133479 | Judith Simcox |
| University of Wisconsin-Madison | WIS04000 | Raghav Jain<br>Judith Simcox |
| NIH Office of the Director | DK128022 | Robert N Helsley |
| University of Kentucky | UL1TR001998 | Robert N Helsley |
| American Heart Association | 19POST34380725 | Iyappan Ramachandiran |

The funders had no role in study design, data collection, and interpretation, or the decision to submit the work for publication.

### Author contributions

Venkateshwari Varadharajan, Conceptualization, Resources, Data curation, Software, Formal analysis, Validation, Investigation, Visualization, Methodology, Writing – original draft, Writing – review and editing; Iyappan Ramachandiran, Rakhee Banerjee, Anthony J Horak, Megan R McMullen, Annette Bellar, Shuhui W Lorkowski, Kailash Gulshan, Isabella James, Nicole Welch, Data curation, Formal analysis, Validation, Investigation, Visualization, Methodology, Writing – review and editing; William J Massey, Data curation, Investigation, Visualization, Methodology, Writing – review and editing; Raghav

Jain, Data curation, Formal analysis, Investigation, Visualization, Methodology, Writing – review and editing; Emily Huang, Data curation, Validation, Investigation, Visualization, Methodology, Writing – review and editing; Robert N Helsley, Vai Pathak, Data curation, Formal analysis, Validation, Investigation, Visualization, Methodology; Jaividhya Dasarathy, Srinivasan Dasarathy, David Streem, Ofer Reizes, Daniela S Allende, Jonathan D Smith, Judith Simcox, Data curation, Formal analysis, Supervision, Validation, Investigation, Visualization, Methodology, Writing – review and editing; Laura E Nagy, Data curation, Formal analysis, Supervision, Validation, Investigation, Visualization, Methodology, Writing – review and editing, helped design experiments and provided useful discussion directing collaborative aspects of the project; J Mark Brown, Conceptualization, Resources, Data curation, Software, Formal analysis, Supervision, Funding acquisition, Validation, Investigation, Visualization, Methodology, Writing – original draft, Project administration, Writing – review and editing

## Author ORCIDs
Venkateshwari Varadharajan https://orcid.org/0000-0002-4557-9840
William J Massey https://orcid.org/0000-0002-2087-6048
Raghav Jain https://orcid.org/0000-0002-1284-6762
Robert N Helsley https://orcid.org/0000-0001-5000-3187
Isabella James https://orcid.org/0009-0000-2796-3497
Srinivasan Dasarathy https://orcid.org/0000-0003-1774-0104
Jonathan D Smith https://orcid.org/0000-0002-0415-386X
Judith Simcox https://orcid.org/0000-0001-7350-6342
J Mark Brown https://orcid.org/0000-0003-2708-7487

## Ethics
The study protocol was approved by the Institutional Review Board for the Protection of Human Subjects in Research at the Cleveland Clinic (IRB 17-718) and MetroHealth Hospital Cleveland (IRB 18-00911). All methods were performed in accordance with the internal review board's guidelines and regulations, and written, informed consent was obtained from all subjects.

All mice were maintained in an Association for the Assessment and Accreditation of Laboratory Animal Care, International-approved animal facility, and all experimental protocols were approved by the Institutional Animal Care and Use Committee of the Cleveland Clinic (IACUC protocols # 2018-2053 and # 00002499).

Reviewer #2 (Public Review): https://doi.org/10.7554/eLife.92243.3.sa1
Author response https://doi.org/10.7554/eLife.92243.3.sa2

---

# Additional files

## Supplementary files
• Supplementary file 1. Primers.
• MDAR checklist

## Data availability
All data generated or analyzed during this study are included in the manuscript and supporting files.

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
