## [Editor Report · eLife assessment]

Varadharajan et al. explore the mechanistic basis of MBOAT7 SNP association with steatotic liver disease and link its function in LPI acylation to altered lipidomics of endosomal/lysosomal system and impaired TFEB mediated lysosomal biogenesis. The findings are **important** with theoretical and practical implications in MAFLD, alcohol-induced hepatic steatosis, and lysosomal diseases. The strength of evidence is **convincing** using methodology in line with current state-of-the-art.

---

## [Referee Report · Reviewer #2 (Public Review)]

Summary:

The work by Varadharajan et. al. explored a previously known genetic variant and its pathophysiology in the development of alcohol-associated liver injury. It provides a plausible mechanism for how varying levels of MBOAT7 could impact the lipid metabolomics of the cell, leading to a deleterious phenotype in MBOAT7 knockout. The authors further characterized the impact of the lipidomic changes and raised lysosomal biogenesis and autophagic flux as mechanisms of how MBOAT7 deletion causes the progression of ALD.

Strengths:

Connecting the GWAS data on MBOAT7 variants with plausible pathophysiology greatly enhances the translational relevance of these findings. The global lipidomic profiling of ALD mice is also very informative and may lead to other discoveries related to lipid handling pathways.

Weaknesses:

The rationale of why MBOAT7 metabolites are lower in heavy drinkers than in normal individuals is not well explained. MBOAT7 loss of function drives ALD, but unclear if MBOAT7 deletion also drives preference for alcohol or if alcohol inhibits MBOAT7 function. Presuming most individuals studied here were WT and expressed an appropriate level of MBOAT7?

Also, discussion of mechanisms of MBOAT7-induced dysregulation of lysosomal biogenesis/autophagy, while very interesting, seems incomplete. It is not clear how MBOAT7 an enzyme involved in membrane phospholipid remodeling increases mTOR which leads to decreased TFEB target gene transcription. Furthermore, given the significant disturbances of global lipidomic profiling in MBOAT7 knockout, many pathways are potentially affected by this deletion. Further in vivo modeling that specifically addresses these pathways (TFEB targeting, mTOR inhibitor) would help strengthen the conclusions of this paper.

---

## [Author Response]

The following is the authors’ response to the original reviews.

**eLife assessment**
The delineation of MBOAT function is important with theoretical and practical implications in MAFLD, alcohol-induced hepatic steatosis, and lysosomal diseases. The strength of evidence is convincing using methodology in line with current state-of-the-art, with good support for the claims.
**Public Reviews:**

**Reviewer #1 (Public Review):**
Summary:The authors provide mechanistic insights into how the loss of function of MBOAT7 promotes alcoholassociated liver disease. They showed that hepatocyte-specific genetic deletion of Mboat7 enhances ethanol-induced hepatic steatosis and increased ALT levels in a murine model of ethanol-induced liver disease. Through lipidomic profiling, they showed that mice with Mboat7 deletion demonstrated augmented ethanol-induced endosomal and lysosomal lipids, together with impaired transcription factor EB (TFEB)-mediated lysosomal biogenesis and accumulation of autophagosomes.Strengths:Alcohol-induced liver disease (ALD) and metabolic-associated steatotic liver disease (MASLD) are major global health problems, and polymorphism near the gene encoding MBOAT7 has been associated with these conditions. This paper is timely as it is important to gain insights on how loss of MBOAT function contributes to liver disease as this may eventually lead to therapeutic strategies. -The conclusions of the paper are mostly well supported by data.

We sincerely thank Reviewer #1 for constructive feedback on this work.

Weaknesses:(1) In regards to circulating levels of MBOAT7 products, a comparison of heavy drinkers with ALD versus heavy drinkers without ALD would be more clinically relevant.

We agree this comparison would be an important comparison to make in future studies, but given the difficulties in accessing well-matched samples such as these we see this as beyond the scope of the current work.

(2) A few typos need to be addressed. For Figure 1 - figure supplement 1, should the second column heading be "Heavy drinkers" instead of "Healthy drinkers"? Also, in the same figure, it is unclear what the "healthy" subcategory under MELD means.

The typographical error was addressed in the main text and in all associated tables and figures.

(3) Some of the data in the tables need to be addressed/discussed. For instance, the white blood cell count (WBC) in Figure 1 - figure supplement 1 for "healthy controls" is 34, compared to 13.51 for drinkers. A WBC of 34 is not at all healthy and should be explained. The vast difference between BMI and also between racial distribution within the two cohorts should also be explained. Is it possible that some of these differences contributed to the different levels of circulating MBOAT7 products that were measured?

Sincere thanks for catching this error. In follow up, we found that some of our patient recruitment sites were using different units to report WBC counts (percent vs 1000/ml) and at this time we cannot retrospectively correct that difference. Therefore, we have incomplete WBC values for the cohort so elected to exclude that information to avoid confusing readers. A revised table is provided in revision reflecting these changes/ If we look at each site separately, values for WBC were in the normal range, so we do not think this is a major limitation of our studies. In regards to BMI and race: Race is not actually significant, but close. For BMI, there are 2 very low BMIs in the Heavy drinkers which bring that average down. We agree with Reviewer # 1 that race and / or BMI could impact MBOAT7, but larger cohorts are needed to detect such potential differences.

(4) The representation of the statistical difference between the bars in the results figures by using alphabets is a bit confusing. For instance, in figure 2C, does that mean all the bars labelled A are significantly different from B? The solid black bar seems to be very similar to the open red bar; please double check.

We apologize for this confusing presentation. Using the letter system, groups not sharing a common superscript differ statistically. Given this concern, we have gone back and reviewed all statistical comparisons and realized that there were several mistakes in the graph Figure 2C, Figure 3F and G, Figure 3-Supplementary Figure 1 F and Figure 3-Supplementary Figure 10H. The graphs themselves were not altered, but the denotation of statistical significance was updated with the correct letter superscripts.

**Reviewer #2 (Public Review):**
Summary:The work by Varadharajan et. al. explored a previously known genetic variant and its pathophysiology in the development of alcohol-associated liver injury. It provides a plausible mechanism for how varying levels of MBOAT7 could impact the lipid metabolomics of the cell, leading to a deleterious phenotype in MBOAT7 knockout. The authors further characterized the impact of the lipidomic changes and raised lysosomal biogenesis and autophagic flux as mechanisms of how MBOAT7 deletion causes the progression of ALD.Strengths:Connecting the GWAS data on MBOAT7 variants with plausible pathophysiology greatly enhances the translational relevance of these findings. The global lipidomic profiling of ALD mice is also very informative and may lead to other discoveries related to lipid handling pathways.

We sincerely thank Reviewer #1 for constructive feedback on this work.

Weaknesses:The rationale of why MBOAT7 metabolites are lower in heavy drinkers than in normal individuals is not well explained. MBOAT7 loss of function drives ALD, but unclear if MBOAT7 deletion also drives preference for alcohol or if alcohol inhibits MBOAT7 function. Presuming most individuals studied here were WT and expressed an appropriate level of MBOAT7?

Although we were unable to genotype for the rs641738 SNP in the human subjects studied here, the original study by Buch et al. published in Nature Genetics performed cis expression quantitative trait lock (cis-eQTL) analyses to demonstrate that the minor disease-associated allele was associated with reduced MBOAT7 expression in subjects with alcohol-related cirrhosis. It is important to note that we did not see any evidence that alcohol preference was altered in either myeloid- or hepatocyte-specific Mboat7-knockout mice, given ethanol intake was similar in all genotypes. Additional studies are needed to address the possibility that MBOAT7 loss of function may promote alcohol preference, but we agree that this should be further investigated.

Also, the discussion of mechanisms of MBOAT7-induced dysregulation of lysosomal biogenesis/autophagy, while very interesting, seems incomplete. It is not clear how MBOAT7 an enzyme involved in membrane phospholipid remodeling increases mTOR which leads to decreased TFEB target gene transcription.

Although we agree with Reviewer #2 that mechanistic understanding by which MBOAT7 loss of function impacts mTOR activity and TFEB-driven lysosomal biogenesis is still incomplete, we do feel that the results published here will inform downstream investigation linking phosphatidylinositol remodeling to mTOR and TFEB. The MBOAT7 gene encodes an acyltransferase enzyme that specifically esterifies arachidonyl-CoA to lysophosphatidylinositol (LPI) to generate the predominant molecular species of phosphatidylinositol (PI) in cell membranes (38:4). It is well established that PI-related lipids can regulate membrane dynamics and signal transduction pathways. For instance PI-phosphates (PIPs) are dynamically shaped by PI kinases and phosphatases to play crucial roles in the regulation of a wide variety of cellular processes via specific interactions of PIP-binding proteins. Among PI phosphates, PI 3phosphate (PI3P) regulates vesicular trafficking pathways, including endocytosis, endosome-toGolgi retrograde transport, autophagy and mTOR signaling. Although additional work is needed to understand the molecular details of how MBOAT7-driven LPI acylation impacts mTOR and TFEB, it is not particularly surprising that PI lipid remodeling could broadly impact cell signaling.

Furthermore, given the significant disturbances of global lipidomic profiling in MBOAT7 knockout, many pathways are potentially affected by this deletion. Further in vivo modeling that specifically addresses these pathways (TFEB targeting, mTOR inhibitor) would help strengthen the conclusions of this paper.

We agree that further in vivo studies are needed that are beyond the scope of the current work.

**Recommendations for the authors:**

**Reviewer #2 (Recommendations For The Authors):**
(1) p values are rather hard to read. For example, Figure 2c, Hepatocyte-specific deletion of Mboat7 resulted in enhanced ethanol-induced increases in liver weight. However, doesn't look like there is a significant difference between the 2 EtOH groups in Figure 2C? Same comment for Figure 2e, not sure if pair-fed groups had a significant difference.(2) Figure 2 Supp fig 1, what is the top band on the MBOAT7 WB?

We have addressed these statistical comparison comments as described above. Although we cannot be sure, it is likely that the top band on the MBOAT7 Western blot is a non-specific band that shows up with the antibody combination used given there is equal intensity in the Mboat7flox/flox and the MSKO mice (Mboat7flox/flox+LysM-Cre).